# ONLINE REINFORCEMENT LEARNING VIA POSTERIOR SAMPLING OF POLICY

## ABSTRACT

We propose a Reward-Weighted Posterior Sampling of Policy (RWPSP) algorithm to tackle the classic trade-off problem between exploration and exploitation under finite Markov decision processes (MDPs). The Thompson sampling method so far has only considered posterior sampling over transition probabilities, which is hard to gain the globally sub-optimal rewards. RWPSP runs posterior sampling over stationary policy distributions instead of transition probabilities, and meanwhile keeps transition probabilities updated. Particularly, we leverage both relevant count functions and reward-weighting to online update the policy posterior, aiming to balance between local and long-term policy distributions for a globally near-optimal game value. Theoretically, we establish a bound of $\tilde{\mathcal{O}}(\Gamma\sqrt{T}/S^2)$[1] on the total regret in time horizon $T$ with $\Gamma/S^2 < D\sqrt{SA}$ satisfied in general, where $S$ and $A$ represents the sizes of state and action spaces, respectively, $D$ the diameter. This matches the best regret bound thus far for MDPs. Experimental results corroborate our theoretical results and show the advantage of our algorithm over baselines in terms of efficiency.

## 1 INTRODUCTION

Online reinforcement learning (Wei et al., 2017) addresses the problem of learning and planning in real-time sequential decision making systems with the interacting environment partially observed or fully observed. The decision maker tries to maximize the cumulative reward during the interaction with the environment, which however inevitably leads to the trade-off between exploration and exploitation. Many attempts have been made to mitigate such dilemma by improving underlying regret bounds (Zhang et al., 2020b)(Ménard et al., 2021)(Zhang et al., 2021b)(Zhang et al., 2022)(Agrawal et al., 2021).

Trade-off between exploration and exploitation has been studied extensively in various scenarios. The goal of exploration is to find as much information as possible of the environment, while the exploitation process aims to maximize the long-term total reward based on the exploited part of the environment. To handle the trade-off problem, one popular way is to use the naive exploration method such as adaptive $\epsilon$-greedy exploration (Tokic, 2010). The method adjusts the exploration parameter adaptively, depending on the temporal-difference (TD) error observed from the value function. Optimistic initialisation methods have also been studied in factored MDPs (Szita & Lörincz, 2009; Brafman & Tennenholtz, 2003). They encourage systematic exploration in the early stage. Another common way is to use the optimism in the face of uncertainty (OFU) principle (Lai & Robbins, 1985), where the agent constructs confidence sets to search for the optimistic parameters associated with the maximum reward. Thompson sampling, as an OFU-based approach, was originally presented for stochastic bandit scenarios (Thompson, 1933). It has been applied in various MDPs contexts (Osband et al., 2013; Agrawal & Goyal, 2012) since it can achieve tighter bounds (Ding et al., 2021; Oh & Iyengar, 2019; Moradipari et al., 2019) and better compatibility with other structures in both theory and practice (Chapelle & Li, 2011; Zhang et al., 2021a; Agrawal & Goyal, 2013). It has also achieved great performance on contextual bandit problems(Agrawal & Jia, 2017)(Osband & Van Roy, 2017)(Osband et al., 2019).The general optimistic algorithms require to solve all MDPs lying within the confident sets, while Thompson sampling-based algorithms only need to solve the sampled MDPs

---

[1]The symbol $\tilde{\mathcal{O}}$ hides logarithmic factors.

to achieve similar results (Russo & Van Roy, 2014). Thompson sampling offers speedup on one hand, and results in biased estimates of the transition matrix on the other hand.

This paper addresses the trade-off problem between exploration and exploitation in finite MDPs. We propose a *reward-weighted posterior sampling of policy (RWPSP)* algorithm that samples posterior policy distributions rather than posterior transition distributions, which optimizes the long-term policy probability distribution. While updating posterior policy distribution, we use the count functions of the state-action pairs to capture the importance of each sampled episode. This way, we manage to optimize the policy distribution in time horizon $T$ and achieve the total regret bound of $\tilde{\mathcal{O}}(\Gamma\sqrt{T}/S^2)$ with $\Gamma/S^2 < D\sqrt{SA}$, where $S$ and $A$ represent the sizes of the state and action spaces respectively. $D$ is the diameter of the finite MDP. In addition, we propose a new Bayesian method to update transition probabilities which also achieves a state-of-art regret bound. In comparison, existing model-based methods like Upper Confidence Stochastic Game Algorithm(*UCSG*) achieve a regret bound of $\tilde{\mathcal{O}}\left(\sqrt[3]{DS^2AT^2}\right)$ on stochastic MDPs (Wei et al., 2017), while model-free methods like *optimistic Q-learning* achieve a regret bound of $\tilde{\mathcal{O}}\left(T^{2/3}\right)$ under infinite-horizon average reward MDPs (Wei et al., 2020). To summarize, this work makes the following contributions:

- We propose a reward-weighted posterior sampling of policy (RWPSP) algorithm that strikes a balance between the posterior projection of the long-term policy and the local policy.

- RWPSP is the first posterior sampling method that samples posterior policy distributions while Bayesian updating transition probabilities. It achieves a regret bound of $\tilde{O}(\frac{\Gamma\sqrt{T}}{S^2})$, where $\Gamma/S^2 < D\sqrt{SA}$. We show that the total regret bound is less than the state-of-the-art, i.e., $D\sqrt{SAT}$, to the best of our knowledge.

- We conduct experimental studies to verify our theoretical results and demonstrate that our RWPSP algorithm outperforms other online learning methods in complex MDP environments.

## 2 RELATED WORK

**Regret Bound Analysis**   In the finite-horizon setting, most of the Thompson Sampling-based algorithms follow a model-based approach (Abbasi Yadkori et al., 2013; Xu & Tewari, 2020; Auer et al., 2008; Fruit et al., 2020; Dong et al., 2020; Agarwal et al., 2020), as model-based reinforcement learning methods are required to approximate the optimal transition matrix of a MDP. In Xu & Tewari (2020), non-episodic factored Markov decision processes are sampled using extreme transition dynamics which encourages visiting new states in order to minimize regret. Although various approaches had been used to minimize the regret bound, current methods still minimize the regret bound by updating the transition matrix. A good comparison can be found in Zhang et al. (2021c); Wei et al. (2020) among existing Thompson sampling based methods. In contrast to existing works with a focus on posterior sampling over transition matrices, our work only considers posterior sampling over policy distributions in a finite-horizon MDP. The transition probabilities will be updated based on the real trajectory. On the other hand, while existing model-free methods have not yet achieved the state-of-art regret bound (Jin et al., 2018; Strehl et al., 2006), some of them improved the total regret bound (Zhang et al., 2020a).

**Intrinsic Reward Shaping**   Intrinsic reward shaping was first introduced in 1999 (Ng et al., 1999), which is a generic idea to guide the policy iteration with intrinsic reward. Count-based methods are then proposed to reach nearly state-of-the-art performance on a high-dimensional environment (Tang et al., 2017). Intrinsic reward is also used in Du et al. (2019) to compute a distinct proxy critic for the agent to guide the update of its individual policy. In order to shape the reward during the policy iteration, we adopt the reward-weighted update to verify the intrinsic reward. Count functions of states and/or actions are usually used in the exploration process of an agent to help build the intrinsic reward (Tang et al., 2017; Bellemare et al., 2016; Burda et al., 2018). In our algorithm, we consider the count function as the posterior projection of the intrinsic reward, and then use the generated reward to update the posterior distribution. The previous methods mainly focus on the instantaneous rewards generated from the exploration process, while our method uses a reward-weighted count function to generate long-term rewards which can guide the policy towards a globally optimal value.

## 3 PROBLEM SETTING

### 3.1 MARKOV DECISION PROCESS

A finite stochastic Markov decision process (fMDP) (Ferns et al., 2004) could be defined by a tuple $M = (\mathcal{S}, \mathcal{A}, r, \theta)$. Denote the sizes of the state and action spaces as $S = |\mathcal{S}|$ and $A = |\mathcal{A}|$, respectively. $r$ represents the reward function defined by $r : S \times A \to [0, 1]^2$. Let $\theta : S \times A \times S$ represent the transition probability such that $\theta (s' \mid s, a) = \mathbb{P}(s_{t+1} = s' \mid s_t = s, a_t = a)$. The ground-truth transition probability $\theta_*$ is randomly generated before the game starts, which is then fixed and unknown to the agent. For the model-based agents, the transition probability at time step $t$ within episode $k$ could be defined as $\theta_{t_k}$. As for each episode, the transition probability would be defined as $\theta_k$. A stationary policy $\pi : S \to A$ is a deterministic function that maps a state to an action. We could define the instantaneous policy under transition probability $\theta_{t_k}$ as $\pi_{\theta_{t_k}}$. The globally optimal policy under local optimal transition probability and global optimal transition probability could be defined as $\pi^*_{\theta_{t_k}}$ and $\pi^*_{\theta_*}$ respectively. For notational brevity, let $\pi_{t_k} \triangleq \pi_{\theta_{t_k}}, \pi^*_{t_k} \triangleq \pi^*_{\theta_{t_k}}, \pi^\star \triangleq \pi^*_{\theta_*}$.

In the fMDP, the average reward function per time step $t$ under stationary policy $\pi$ is defined as:

$$J(\pi_{\theta_t}) = \lim_{T \to \infty} \frac{1}{T} \mathbb{E} \left[ \sum_{t'=t}^{t+T} r(s_{t'}, a_{t'}) \right]. \tag{1}$$

Therefore, we could denote the instantaneous average reward under transition probability $\theta_{t_k}$ as $J(\pi_{t_k})$. Note that $J(\pi_{t_k})$ is a hypothetical average reward generated from $\theta_{t_k}$ and $\pi_{t_k}$. The locally optimal average reward $J(\pi^*_{t_k})$ could be derived from the corresponding locally optimal policy $\pi^*_{t_k}$. The globally optimal average reward could be represented as $J(\pi^\star)$. Define the maximum average reward as $\Gamma = \max J(\pi_\theta)$, which is the maximum average reward that an agent could achieve during its exploration in a fMDP. The maximum value $\Gamma$ will be achieved under the optimal transition probability with the optimal stationary policy, i.e., $\Gamma = J(\pi^\star)$.

In the online learning setting, total regret is defined to be the difference between the optimal total game value and the actual game value as follows:

$$Reg = \max_a \sum_{t=1}^T r(s_t, a) - \sum_{t=1}^T r(s_t, a_t). \tag{2}$$

It is used to measure the performance of a decision maker. Since this metric is hard to calculate in general, we define the following *bias vector* $b(\theta, \pi, s)$ (Wei et al., 2017) as the relative advantage of each state to help us measure the total regret.

$$b(\theta, \pi, s) \triangleq E \left[ \sum_{t=1}^\infty r(s_t, a_t) - J(\pi) \mid s_1 = s, a_t \sim \pi(\cdot | s_t) \right]. \tag{3}$$

Under stationary policy $\pi$, the advantage of one state $s$ over another state $s'$ is defined as the difference between their accumulated rewards with initial states as $s$ and $s'$, respectively, which will eventually converge to the difference of their bias vectors, i.e., $b(\theta, \pi, s) - b(\theta, \pi, s')$. Denote the expected total reward under stationary policy $\pi$ by $r(s, \pi) = E_{a \sim \pi(\cdot | s)}[\sum r(s, a)]$, and the expected transition probability by $p_\theta(s' \mid s, \pi) = E_{a \sim \pi(\cdot | s)}[p_\theta(s' | s, a)]$. The bias vector then satisfies the Bellman equation below:

$$J(\pi_\theta) + b(\theta, \pi, s) = r(s, \pi) + \sum_{s'} p_\theta(s' \mid s, \pi) b(\theta, \pi, s'). \tag{4}$$

Define the *span* of a vector $x$ as $\mathrm{sp}(x) = \max_i x_i - \min_i x_i$. The regret is strongly connected to the span of bias vector $b(\theta_*, \pi^\star, \cdot)$, i.e., $\mathrm{sp}(b(\theta_*, \pi^\star, \cdot))$. The span of any $b(\theta, \pi, \cdot)$ is upper bounded by $D \triangleq \max T^{\pi_{t_k}}_{s \to s'}(\theta_{t_k})$, i.e., the maximum expected time to reach to state $s'$ from state $s$ under transition probability $\theta_{t_k}$ and policy $\pi_{t_k}$.

---

[2]In a finite MDP, the reward in each episode should be confined within $[0, 1]$.

## 3.2 Assumptions

The globally optimal policy is hard to learn for MDPs under online settings. As they often get stuck in locally optimal results. The $\epsilon$-tolerance is then introduced to help measure the performance of the algorithm. When the difference between the current average reward and the optimal average reward is less than constant $\epsilon$, the current policy is said to be $\epsilon$-optimal.

**Assumption 3.1.** *($\epsilon$-Optimal policy) (Hartman, 1975) Under sub-optimal and optimal transition probability, if policy $\pi_{t_k}$ satisfies*

$$J_{\pi_*}(\theta_{t_k}) - J_{\pi_{t_k}}(\theta_{t_k}) \leq \epsilon.$$

*Then, policy $\pi_{t_k}$ is $\epsilon$-optimal.*

Assumption 3.2 implies that under all circumstances, all the states could be visited in $D$ steps on average. When the agent conducts the optimal policy under the optimal transition probability, the transition time $T_{s \to s'}^{\pi^*}(\theta_*)$ should be the shortest, because the agent tends to explore the fewest non-related states under the optimal stationary policy. In a similar fashion, the transition time $T_{s \to s'}^{\pi_{t_k}^*}(\theta_{t_k})$ for agent conducting optimal policy under sub-optimal transition probability is assumed to be less than the maximum transition time $D = \max_{s,s'} T_{s \to s'}^{\pi_{t_k}}(\theta_{t_k})$ in normal settings.

**Assumption 3.2.** *(Expected transition time) When conducting stationary policy $\pi$, we assume that the maximum expected time to reach to state $s'$ from state $s$ under sub-optimal transition probability and optimal transition probability is less than constant $D$:*

$$\max T_{s \to s'}^{\pi^*}(\theta_*) \leq \max T_{s \to s'}^{\pi_{t_k}^*}(\theta_{t_k}) \leq \max T_{s \to s'}^{\pi_{t_k}}(\theta_{t_k}) = D.$$

Let $e(t) \triangleq k$ be the episode where time instant $t$ belongs. When conducting stationary policy $\pi$, we could define the count function for the episode number $e(t)$ as $N(\pi_{e(t)})$. Define $\mathcal{H}_{s_1, s_2}(k, \pi)$ as the set of all the time instants that the state transition $s_1 \to s_2$ occurs in the first $k$ episodes with stationary policy $\pi$ used:

$$\mathcal{H}_{(s_1, s_2)}(k, \pi) \triangleq \sum_{t=1}^{\infty} \mathbb{1}\left\{\pi_{e(t)} = \pi, (S_t, S_{t+1}) = (s_1, s_2), N(\pi_{e(t)}) \leq k\right\}. \tag{5}$$

Under transition probability $\theta_{t_k}$, the expected transition time from state $s$ to state $s'$ with stationary policy $\pi_{t_k}$ could be denoted as $\tilde{\tau}_{\pi_{t_k}}$, i.e., $\tilde{\tau}_{\pi_{t_k}} \triangleq T_{s \to s}^{\pi_{t_k}}(\theta_{t_k})$. Therefore, the posterior probability of the stationary policy $\pi$ can be assumed as the difference between the empirical state pair frequency $\frac{\mathcal{H}_{(s_1, s_2)}(k, \pi)}{k}$ and the corresponding expected value $\tilde{\tau}_{\pi_{t_k}}$.

**Assumption 3.3.** *(Posterior distribution under sub-optimal trajectories) (Gopalan & Mannor, 2015) For any given $e_1, e_2 \geq 0$, there exists $p \triangleq p(e_1, e_2) > 0$ satisfying $\theta_{t_k}(\pi_{t_k}^*) \geq p$ for any episode index $k$ at which sub-optimal transition frequencies have been observed:*

$$\left|\frac{\mathcal{H}_{(s_1, s_2)}(k, \pi)}{k} - \tilde{\tau}_{\pi_{t_k}}\theta(s_1|s_2)\right| \leq \sqrt{\frac{e_1 \log(e_2 \log k)}{k}}, \quad \forall s_1, s_2 \in \mathcal{S}, k \geq 1.$$

## 4 Proposed Algorithms

In this section, we propose a new algorithm to tackle the trade-off between exploration and exploitation. One parameter that we need under the posterior sampling setting is the prior distribution, denoted as $\mu_0$. Note that we generate prior distributions for both transition probabilities and stationary policies, but only do posterior sampling over stationary policy distributions. While the transition probabilities will be Bayesian updated by the trajectory generated from the posterior policy. In each episode $k$, at each time step $t$, the action would be sampled from the posterior policy distribution. And such policy distribution $\mu_{t_k}(\pi)$ will be updated based on the previous history $h_{t_k}$. Let $N_t(s, a)$ be the number of visits to any state-action pair $(s, a)$ during a period of time $t$:

$$N_t(s, a) = |\{\tau < t : (s_\tau, a_\tau) = (s, a)\}|. \tag{6}$$

---

**Algorithm 1** Reward-Weighted Posterior Sampling of Policy (RWPSP)

---

**Input**: Game environment, prior distribution for stationary policy $\mu_{\pi_0}$, transition probability $\theta_0$, initial state $s_0 \in S$, time step $t = 0$.
**Output**: Stationary policy $\pi_K$

 1: **for** Episode $k = 0, 1, 2 \ldots K$ **do**
 2:     $T_{k-1} \leftarrow t - t_k$
 3:     $t_k \leftarrow t$
 4:     Generate $\mu_k(\pi_k)$ based on prior distribution
 5:     Update $\theta_k$ using $\theta_k = \frac{\theta_{k-1}(s_1|s_2,a_2) + H_{s_1,s_2}(N_{\pi_t}(k), \pi)}{N_{\pi_t}(k)}$
 6:     **for** $t \leq t_k + T_{k-1}$ and $N_t(s, a) \leq 2N_{t_k}(s, a)$ **do**
 7:         Sample $\pi_{t_k} \sim \mu_{t_k}(\pi)$ and apply action $a_t \sim \pi_{t_k}$
 8:         Observe new state $s_{t+1}$, reward $r_{t+1}$
 9:         Update posterior distribution $\mu_{(t+1)_k}(\pi)$ using RWPI
10:         $t \leftarrow t + 1$
11:     **end for**
12: **end for**

---

We then have our algorithm called the Reward-Weighted Posterior Sampling of Policy, RWPSP for short, described in Algorithm 1.

At the beginning of each episode $k$, the RWPSP algorithm, i.e., Algorithm 1, samples a policy distribution from the prior distribution $\mu_{(t-1)_k}(\pi_{k-1})$ (Line 4), which equals the updated posterior policy distribution from the last episode (Line 9). Then, the transition probability distribution will be generated from the history transition matrix $\theta_{k-1}$ and count function $H_{s_1,s_2}(N_{\pi_t}(k), \pi)$ and $N_{\pi_t}(k)$ (will be defined in Section 4.1) (Line 5). We use two stopping criteria to limit agent's exploration direction. The first stopping criterion aims to stop meaningless exploration, while the second stopping criterion ensures that any state-action pair $(s, a)$ will not be encountered twice during the same episode (Line 6). At each time step $t_k$, actions are generated from the instantaneous policy $\pi_{t_k}$ (Line 7) which follows a posterior distribution $\mu_{t_k}(\pi)$. These actions are then be used by the agent to interact with the environment to observe the next state $s_{t+1}$ and the reward $r_{t+1}$ (Line 8). The observation results are then be used to find the optimal posterior distribution for policy $\pi_{(t+1)_k}$ (Line 9).

### 4.1 UPDATE RULE

In previous Bayesian methods, the transition matrix is updated with Thompson/posterior sampling. But in our case, we apply posterior sampling over the policy distributions. Based on the Bayes' rule, the posterior distribution of policy can be written as :

$$\mu_{t+1}(\pi) = \frac{\theta(s_{t+1} \mid s_t, a_t) \mu_{t_k}(\pi)}{\sum_{\pi'} \theta'(s_{t+1} \mid s_t, a_t) \mu_t(\pi')}. \tag{7}$$

The way we update stationary policy resembles how Thompson sampling updates transition probabilities, as our algorithm uses the prior policy to guide the current policy. The key difference is that our Reward-Weighted Policy Iteration (RWPI) algorithm shown in Algorithm 2 is able to balance between the instantaneous action and the history actions. This will help our method approximate the long-term maximum reward, which is the globally optimal value in this scenario. We could define $W_{t_k}$ as the *posterior weight* in episode $k$ at time $t$ (Line 2 in Algorithm 2). Let $J_{\pi_{t_k}}$ and $J_{\pi_{t_k}^*}$ denote the instantaneous average reward and the locally optimal value. This locally optimal value is induced by adopting the greedy policy on the transition probabilities $\theta_t$. The value of $W_{t_k}$ is proportional to the log difference between the average reward of the locally optimal policy and the current policy. At last, we could generate the policy distribution $\mu_t(\pi)$ based on the previous policy distribution $\mu_{t-1}(\pi)$ and the current locally optimal policy $\pi_t^*(s, \theta_t)$ (Line 3 in Algorithm 2).

We measure the distance between the history optimal policy and the instantaneous policy using the *Marginal Kullback-Leibler Divergence* (Marginal KL Divergence) which is a widely used metric

---

**Algorithm 2** Reward-Weighted Policy Iteration(RWPI)

---

**Input**: Game environment, prior distribution for stationary policy $\mu_t(\pi)$
**Output**: Stationary policy $\pi_i$

1: **repeat**
2:     $W_{t_k}(\pi) = \exp\{\sum_{\pi,s} \mathcal{H}_s(N_\pi(k), \pi) \log \frac{J_{\pi_t}}{J_{\pi^*}(s)}\}$
3:     $\mu_t(\pi) = W_t \mu_{t-1}(\pi) + (1 - W_t)\pi_t^*(s, \theta_t)$
4: **until** $D_\pi(\mu_*(\pi)||\mu_{t_k}(\pi)) \leq \epsilon$

---

for characterizing the difference between two probability distributions. The distance then could be written as follows:

$$
\begin{aligned}
D_\pi(\mu_*(\pi)||\mu_{t_k}(\pi)) &\triangleq \sum_{s_1 \in \mathcal{S}} \theta_{s_1}^\pi \sum_{s_2 \in \mathcal{S}} \mu_*(\pi) \log \frac{\mu_*(\pi)}{\mu_{t_k}(\pi)} \\
&= \sum_{s_1 \in \mathcal{S}} \theta_{s_1}^\pi \mathbb{KL}\left(\mu_*(\pi)||\mu_{t_k}(\pi)\right).
\end{aligned}
$$

Parameter $\epsilon$ in Algorithm 2 represents the tolerance between the optimal policy and the instantaneous policy. RWPI updates the policy dynamically with the posterior weight. The policy will converge to an $\epsilon$-optimal value after certain number of iterations under this update method. In the following section, we will analyze the convergence of this posterior update method and the total regret bound of our method.

## 5 MATHEMATICAL ANALYSIS

### 5.1 CONVERGENCE OF THE UPDATE RULE

We show the convergence of our posterior policy update method to demonstrate its superiority. To this end, we need the following three Lemmas. Lemma 5.1 shows that RWPI enjoys asymptotic convergence. We then demonstrate in Lemma 5.2 that the output policy of such policy iteration method updates monotonically towards the optimal direction, which is vital evidence for the global optimality of our update method. At last, Lemma 5.3 proves that under MDP $M$, the output policy generated from the RWPI method would reach $\epsilon$-optimality after a constant number of iterations.

**Lemma 5.1.** *Suppose Assumption 3.2 holds for some stochastic MDP $M$, then the policy iteration algorithm on $M$ converges asymptotically.*

*Proof.* If Assumption 3.2 holds, by Theorem 4 in Wal, van der (1977), the successive policy approximation process yields an $\epsilon$-band and stationary $\epsilon$-optimal strategies for the agent. This results match Assumption 3.1. Therefore, the convergence of the policy could be proved. □

**Lemma 5.2.** *The average reward deducted by Algorithm 2 will be monotonically increasing.*

*Proof.* From Algorithm 2, we can write the update rule of the average reward as follows:

$$
\begin{aligned}
J_{\pi_t}(\theta) - J_{\pi_{t-1}}(\theta) &= (W_t - 1)J_{t-1}(\theta) + (1 - W_t)J_{\pi^*}(s, \theta) \\
&= (1 - W_t)(J_{\pi^*}(s, \theta) - J_{\pi_{t-1}}(\theta)).
\end{aligned}
\tag{8}
$$

If $J_{\pi_{t-1}}(\theta) \leq J_{\pi^*}(s, \theta)$, then $W_t \leq 1$ since $\log \frac{J_{\pi_t}(\theta)}{J_{\pi^*}(s,\theta)} \leq 0$, otherwise $W_t \geq 1$. Thus,

$$
J_{\pi_t}(\theta) - J_{\pi_{t-1}}(\theta) = (1 - W_t)(J_{\pi^*}(s, \theta) - J_{\pi_{t-1}}(\theta)) \geq 0.
\tag{9}
$$

That is, the sequence $J_{\pi_t}(\theta)$ is monotonically increasing with time step $t$. □

**Lemma 5.3.** *Suppose Assumptions 3.1-3.2 hold for some stochastic MDP $M$. Let $u_i$ be the state value in iteration $i$. Define $N$ as the maximum iteration number of the algorithm. Then $\pi_{t_k}$ is $\epsilon$-optimal after $N$ iterations.*

*Proof.* The detailed proof is shown in Appendix A.2 □

## 5.2 REGRET BOUND ANALYSIS

In the proof of the regret bound, we always consider for the worst case. The randomness of the algorithm is reduced the minimum level in order to get fair measurement of the performance of the algorithms. After proving the convergence of the RWPI method, we now turn to the proof of the total regret bound. The regret in time horizon $T$ can be written as:

$$
\begin{aligned}
Reg_T &= T J_{\pi_K}(\theta_K) - \sum_{t=1}^{T} r_{\pi_t}(s_t, a_t) \\
&\approx T J_{\pi_K}(\theta_K) - \sum_{t=1}^{T} J_{\pi_t}(\theta_K) + \sum_{t=1}^{T} J_{\pi_t}(\theta_K) - \sum_{t=1}^{T} J_{\pi_t}(\theta_t) \\
&= Reg_T^1 + Reg_T^2.
\end{aligned}
\tag{10}
$$

Let the episode number be $K$ in time horizon $T$. The regret are defined separately as $Reg_T^1 = T J_{\pi_K}(\theta_K) - \sum_{t=1}^{T} J_{\pi_t}(\theta_K)$ and $Reg_T^2 = \sum_{t=1}^{T} J_{\pi_t}(\theta_K) - \sum_{t=1}^{T} J_{\pi_t}(\theta_t)$. The final average reward under the final policy $\pi_k$ and final transition matrix $\theta_K$ is defined as $J_{\pi_K}(\theta_K)$. $Reg_T^1$ represents the posterior policy regret and $Reg_T^2$ represents the posterior transition probability regret. For any measurable function f and any $h_{t_k}$-measurable random variable $X$, $\mathbb{E}\left[f\left(\theta_*, X\right) \mid h_{t_k}\right] = \mathbb{E}\left[f\left(\theta_k, X\right) \mid h_{t_k}\right]$ (Osband et al., 2013).

In order to bound the first regret $Reg_T^1$, we first bound the ratio between the expected optimal average reward and the instantaneous reward. Based on Assumptions 3.1-3.3, the expected optimal reward that an agent could achieve in the fMDP could be bounded by parameter $\Gamma$ and $\epsilon$.

**Lemma 5.4.** $\log \frac{J_{\pi_*}(\theta)}{J_{\pi_t}(\theta)} \leq \frac{\epsilon}{\Gamma}$.

*Proof.* The detailed proof is shown in Appendix A.2 □

After bounding the log ratio between the expected optimal average reward and the instantaneous reward, we now bound the instantaneous posterior weight $W_{t_k}$, which is important to our proof. At each time step, the posterior weight will be updated based on the previous policy and the observed data. First, we define the counter function $N_\pi(t) := \sum_{t=0}^{t-1} \sum_\pi \mathbb{1}\left\{\pi_{e(t)} = \pi\right\}$ as the total number of the time instants during the period of $t$ when policy $\pi$ was conducted. When Assumption 3.3 holds, we could bound the posterior weight based on the count function in episode $k$ and the average transition time $\tilde{\tau}$.

**Lemma 5.5.** *Under Assumption 3.3, for each stationary near-optimal policy $\pi$ and episode $k \geq 1$. The following upper bound holds for negative log-density:*

$$
-\log W_{t_k}(\pi) \leq \frac{\epsilon}{\Gamma}|S|^2 (\rho(k_\pi)\sqrt{k_\pi} + k_\pi \tilde{\tau}_{t_k, k_\pi}).
$$

The real reward is expected to get close to the expected reward by certain optimization method. A large number of iterations would be needed for this purpose. Therefore, from the convergence proof we proposed in section 5.1, we could derive the bound on the expected convergence time during the optimization process. In Lemma 5.6, we give the bound on the instantaneous difference between the real reward and the expected reward with $\sqrt{T}$. This bound is inversely proportional to $\sqrt{T}$, since our update method updates towards the optimal direction (see Lemma 5.2). For brevity, the full proof will be given in Appendix A.5

**Lemma 5.6.** *The difference between the local optimal average reward and the instantaneous average reward can be bounded as $|J_{\pi_t} - J^*| \leq \tilde{\mathcal{O}}(\frac{\Gamma}{S^2 \sqrt{T}})$.*

We then could combine the previous Lemmas together to get the final regret bound of $Reg_T$.

**Theorem 5.7.** *The first part of the regret in time horizon $T$ is bounded by: $Reg_T \leq \tilde{\mathcal{O}}(\frac{\Gamma\sqrt{T}}{S^2})$.*

It is not clear if the above result improves over the state-of-art methods. We further give a tighter bound for our method below, which shows that under the fMDP our method has a lower regret bound compared to the current state-of-art method $\tilde{O}(D\sqrt{SAT})$.

**Lemma 5.8.** $\frac{\Gamma}{S^2} < D\sqrt{SA}$ when $|S| \geq 2$ or $|A| \geq 2$.

The second regret $Reg_T^2$ represents the posterior difference generated by the update method of the transition probability. First, we use the definition of the Bellman iterator of the average reward to transmit the one-step posterior transition difference into the difference between the transition probability. Then we apply the Assumption 3.3 to help bound such difference. At last, the regret could be bounded by summing all the one-step posterior transition difference.

**Theorem 5.9.** *The regret caused by transition matrix update could be bounded by:*

$$Reg_T^2 \leq \tilde{O}(D(SAT)^{\frac{1}{4}}).$$

## 6 EXPERIMENT

In this section, we compare our method with various state-of-the-art methods: *SACL* (Fruit et al., 2018a), *UCRL2* (Auer et al., 2008), *UCRL2B* (Fruit et al., 2020), *UCRL3* (Bourel et al., 2020), and *KL-UCRL* (Talebi & Maillard, 2018). *SACL* is an exploartion-based method that uses a proper exploration bonus to solve any discrete unknown weakly-communicating MDP. It admits a regret bound $\mathcal{O}\left(D\sqrt{\sum_{s,a} K_{s,a} T \log(T/\delta)}\right)$ (Fruit et al., 2018b). *UCRL2*, *UCRL2B*, and *UCRL3* are three optimistic methods that used certain confidence bounds to minimize the total regret. The *UCRL2* algorithm performs the regret minimization in unknown discrete MDPs under average-reward criterion. *UCRL2B* refines the previous *UCRL2* method by exploiting empirical Bernstein inequalities to prove a regret bound of $\widetilde{\mathcal{O}}(D\sqrt{\Gamma SAT})$ where $\max_{s,a} \Gamma(s,a) \leq S$. *UCRL3* modifies the previous algorithms by using time-uniform concentration inequalities to compute confidence sets on the reward and transition distributions for each state-action pair. Finally, the *KL-UCRL* studied the ergodic MDPs and proposed a high-probability regret bound $\widetilde{\mathcal{O}}\left(\sqrt{S \sum_{s,a} \mathbf{V}_{s,a}^\star T}\right)$, where $\mathbf{V}_{s,a}^\star$ is the variance of the bias function with respect to the next-state distribution following action $a$ in state $s$.

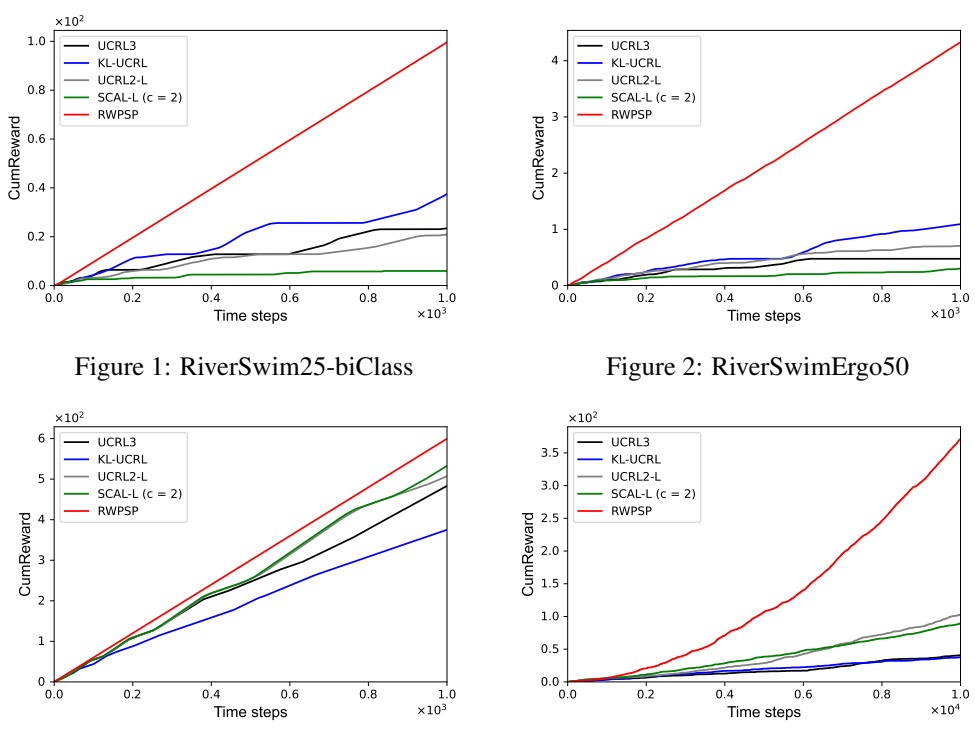

Figure 1: RiverSwim25-biClass

Figure 2: RiverSwimErgo50

Figure 3: Three-States

Figure 4: Four-Rooms

In order to measure the performance of our method empirically, we consider several traditional game environments: RiverSwim, 4-room, and three-state. RiverSwim is one of the most important

metrics for online learning algorithms. It was first proposed in Strehl & Littman (2008) by Michael L. Littman in 2008. The basic RiverSwim consists of six states. The agent starts from a random state and the two actions available to the agent are to swim left or right. But the current will push the agent to the left side. The agent will receive a much larger reward for swimming upstream and reaching the rightmost state. In our experiment, we use its enhanced version: RiverSwim25-Biclass and RiverSwimErgo50. The RiverSwim25-Biclass is a 25-state communicating riverSwim environment with transition probability for the middle states cut in two subsets. And the RiverSwimErgo50 is a 50-states ergodic RiverSwim environment. For the three-state environment, it was first proposed in Fruit et al. (2018b) as the metric for the *SACL*. It is an environment with random reward that contains three states and two actions. 4-room is a classic reinforcement learning environment, where the agent must navigate in a maze composed of four rooms interconnected by 4 gaps in the walls. To obtain a reward, the agent must reach the green goal square.

In the experiment, we use the cumulative reward as the metric. We can see from Figure 1 and Figure 2 that the RWPSP method tends to perform better in the high-dimensional games like RiverSwim25-biclass and RiverSwim-Ergo50 than other state-of-the-art methods, which matches our theoretical analysis since RWPSP is designed to discover the long-term average reward in finite-horizon MDPs. Also, our algorithm performs well in three-state case and surpasses the performance of *SCAL*. We observe that our method RWPSP shows significant improvements over other methods in RiverSwim25-Biclass, RiverSwimErgo50 and 4-room. That is because the total regret bound $\tilde{O}(\frac{\Gamma\sqrt{T}}{S^2})$ of our method indicates that the regret bound will decrease when the number of states of the environment increases. Thus, our method RWPSP performs pretty well on complex online learning environments.

## 7 CONCLUSION

In this work, we propose a policy-based posterior sampling method that can achieve the best total regret bound $\tilde{\mathcal{O}}(\Gamma\sqrt{T}/S^2)$ in finite-horizon stochastic MDPs. This algorithm provides a new way to trade-off between exploration and exploitation by sampling from the posterior distributions of policy. The posterior policy can be updated to balance between the long-term policy and the current greedy policy. Our study shows that this posterior sampling method outperforms other optimization algorithms both theoretically and empirically.

Despite that the sampling method is known to be efficient in discrete environments, our work shows that it could be further improved with count functions and reward re-weighting for posterior updates. However, it remains unknown in this work if similar ideas are applicable to continuous environments as well, which we leave to our future work. For example, we may use some metric to accommodate the difference between states of a continuous space, and then apply our algorithm to such environments.

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

# A    DETAILS OF PROOFS

The appendix aims to introduce the complete proof of the previous lemmas and theorems.

## A.1    THE CONVERGENCE OF PI

**Lemma A.1.** *Under update algorithm RWPI, the average reward should be monotonically increased.*

*Proof.* From Algorithm 2, we could deduce the update rule of the average reward:

$$
\begin{aligned}
J_{\pi_t}(\theta) - J_{\pi_{t-1}}(\theta) &= (W_t - 1)J_{t-1}(\theta) + (1 - W_t)J_{\pi^*}(s, \theta) \\
&= (1 - W_t)(J_{\pi^*}(s, \theta) - J_{\pi_{t-1}}(\theta)).
\end{aligned}
\tag{11}
$$

When $J_{\pi^*}(s, \theta) \geq J_{\pi_{t-1}}(\theta)$, we could deduce that $\log \frac{J_{\pi_t}(\theta)}{J_{\pi^*}(s,\theta)} \leq 1$. So the posterior weight $W_t$ is less than 1. This result holds vice versa. The first term $1 - W_t \leq 0$ when $J_{\pi^*}(s, \theta) \leq J_{\pi_{t-1}}(\theta)$. Therefore, we could prove that:

$$
J_{\pi_t}(\theta) - J_{\pi_{t-1}}(\theta) = (1 - W_t)(J_{\pi^*}(s, \theta) - J_{\pi_{t-1}}(\theta)) \geq 0.
\tag{12}
$$

The sequence $J_{\pi_t}(\theta)$ is monotonically increased with time step $t$.    □

**Lemma A.2.** *Suppose Assumption 1 and Assumption 2 hold for some stochastic MDP $M$. Let $u_i$ be the state value in iteration $i$. Define $N$ as the maximum iteration number of the algorithm. Then $\pi_{t_k}$ is $\epsilon$-optimal after $N$ iterations.*

*Proof.* Define $D = \min_s\{\mu_{i+1}(\pi) - \mu_\pi\}$ and $U = \max_s\{\mu_{i+1}(\pi) - \mu_i(\pi)\}$. Then we could deduce:

$$
\begin{aligned}
D + \mu_N(\pi) &\leq \mu_{N+1} \\
&\leq W_i\mu_N + (1 - W_i)\pi_i^*(s, \theta) \\
&\leq W_i\mu_N + (1 - W_i)(r_N + \theta u_N).
\end{aligned}
\tag{13}
$$

Since $0 < W_i \leq 1$, the upper equation could be turned to:

$$
D \leq (1 - W_i)J_{\pi_i}(\theta).
\tag{14}
$$

Based on the definition in Preliminaries, let $\pi^\star$ be the optimal policy under all states that satisfies $\pi^\star := \sum_{s \in S} \pi_i^\star(s, \theta)$. Then

$$
D \leq (1 - W_i)J_{\pi_i}(\theta) \leq (1 - W_i)J_{\pi^\star}(\theta).
\tag{15}
$$

In a similar way, we could also prove $U \geq (1 - W_i)J_{\pi^\star}(\theta)$. From the definition of the stopping criterion of the Policy Iteration algorithm, we could assume $U - D \leq (1 - W_i)\epsilon$. Therefore, we have

$$
\begin{aligned}
U &\leq D + (1 - W_i) \\
&\leq (1 - W_i)J_{\pi_i}(\theta) + (1 - W_i)\epsilon \\
&\leq (1 - W_i)(J_{\pi_i}(\theta) + \epsilon) \\
(1 - W_i)J_{\pi^\star} &\leq (1 - W_i)(J_{\pi_i}(\theta) + \epsilon) \\
J_{\pi^\star} &\leq J_{\pi_i}(\theta) + \epsilon.
\end{aligned}
\tag{16}
$$

We could deduce that stationary policy $\pi$ is $\epsilon$-optimal after $N$ iterations.    □

## A.2    REGRET BOUND ANALYSIS

**Lemma A.3.**
$$
\log \frac{J_{\pi_*}(\theta)}{J_{\pi_t}(\theta)} \leq \frac{\epsilon}{\Gamma}
$$

*Proof.* First, we could multiply $J_{\pi_t}(\theta)$ in order to construct the inequality. Let $J_{\pi_t}(\theta) = n, \epsilon = x$

$$
\begin{aligned}
\lim_{n \to +\infty} \left(1 + \frac{x}{n}\right)^n &= \lim_{n \to +\infty} e^{n \ln\left(1 + \frac{x}{n}\right)} \\
&= e^{\lim_{n \to +\infty} \frac{\ln\left(1 + \frac{x}{n}\right)}{\frac{1}{n}}}.
\end{aligned}
\tag{17}
$$

Apply the L'Hopital's Rule:

$$
\lim_{n \to +\infty} \left(1 + \frac{x}{n}\right)^n = e^{\lim_{n \to +\infty} \frac{\left(\frac{-x}{n^2}\right)\frac{1}{1+\frac{x}{n}}}{-\frac{1}{n^2}}} \tag{18}
$$

$$
= e^{\lim_{n \to +\infty} \frac{x}{1+\frac{x}{n}}} = e^x.
$$

Then, we could prove that $\left(1 + \frac{x}{n}\right)^n$ is monotonically increased with $n$:

$$
\begin{aligned}
(1 + \frac{x}{n})^2 &= 1 \cdot \underbrace{\left(1 + \frac{x}{n}\right) \cdot \left(1 + \frac{x}{n}\right) \cdots \cdots \left(1 + \frac{x}{n}\right)}_{n} \\
&\leq \left[\frac{1 + (1 + \frac{x}{n}) + \cdots + (1 + \frac{x}{n})}{n+1}\right]^{n+1} \\
&= \left[\frac{1 + n(1 + \frac{x}{n})}{n+1}\right]^{n+1} \\
&= \left[1 + \frac{x}{n(n+1)}\right]^{n+1} \\
&\leq \left[1 + \frac{x}{n+1}\right]^{n+1}.
\end{aligned} \tag{19}
$$

The first inequality holds for the arithmetic mean equality. We could deduce that $\left(1 + \frac{x}{n}\right)^n \leq e^x$. Therefore, we have:

$$
J_{\pi_t}(\theta) \log \frac{J_{\pi_*}(\theta)}{J_{\pi_t}(\theta)} \leq \epsilon. \tag{20}
$$

Based on the definition of $\Gamma$, we could deduce the upper bound of average reward. Then the lemma could be proved. $\qquad\square$

**Lemma A.4.** *Under Assumption 3, for each stationary near-optimal policy $\pi$ and epoch counter $k \geq 1$. Let $\rho(x)$ satisfies $\rho(x) := O(\sqrt{\log(\log(x))})$. The following upper bound holds for negative log-density.*

$$
-\log W_{t_k}(\pi) \leq \frac{\epsilon}{\Gamma}|S|^2(\rho(k_\pi)\sqrt{k_\pi} + k_\pi \tilde{\tau}_{t_k, k_\pi})
$$

*Proof.* When $W_{t_k} \leq 1$, we could have:

$$
W_{t_k}(\theta) := \exp \sum_{\pi, s_1, s_2} \mathcal{H}\left(N_\pi(k), \pi\right) \log \frac{J_{\pi_t}(\theta)}{J_{\pi_*}(\theta)}. \tag{21}
$$

Based on the definition of the counter $\mathcal{H}$, we could deduce the value of the posterior weight in a single epoch:

$$
\begin{aligned}
W_{t_k}(\theta) &= \exp\left(\sum_{t=1}^{\infty} \mathbb{1}\left\{\pi_{e(t)} = \pi, (S_t, S_{t+1}) = (s_1, s_2), N(e(t)) \leq k\right\} \log \frac{J_{\pi_t}(\theta)}{J_{\pi_*}(\theta)}\right) \\
&= \exp\left(\sum_{\pi \in \Pi} \sum_{(s_1, s_2) \in \mathcal{S}^2} \sum_{t=1}^{T} \mathbb{1}\left\{\pi_{e(i)} = \pi, (S_t, S_{t+1}) = (s_1, s_2)\right\} \log \frac{J_{\pi_t}(\theta)}{J_{\pi_*}(\theta)}\right) \\
&= \exp\left(N_\pi(t) \sum_{(s_1, s_2) \in \mathcal{S}^2} \sum_{t=0}^{t-1} \frac{\mathbb{1}\left\{\pi_{e(t)} = \pi, (S_t, S_{t+1}) = (s_1, s_2)\right\}}{N_\pi(t)} \log \frac{J_{\pi_t}(\theta)}{J_{\pi_*}(\theta)}\right).
\end{aligned} \tag{22}
$$

Where $N_\pi(t) := \sum_{t=0}^{t-1} \sum_{\pi \in \Pi} \mathbb{1}\left\{\pi_{e(t)} = \pi\right\}$ represents the total number of the time instants during the period of $t$ when policy $\pi$ was conducted.

When Assumption 3 holds, we could know that $N_\pi(t) = \tilde{\tau}_{\pi_{t_k}, N_\pi(k)}$, where $N_\pi(k) := \sum_{k=0}^K \sum_{\pi \in \Pi} \mathbb{1}\{\pi_{e(k)} = \pi\}$ holds for the number of the epochs where policy $\pi$ was chosen(The notation of $\tau$ will be represented as $N_\pi(k) = k_\pi$, $\tilde{\tau}_{\pi_{t_k}, N_\pi(k)} = \tilde{\tau}_{t_k, k_\pi}$). Therefore, we could have:

$$- \log W_{t_k}(\pi)$$

$$= -N_\pi(t) \sum_{(s_1, s_2) \in \mathcal{S}^2} \sum_{t=0}^{t-1} \frac{\mathbb{1}\{\pi_{e(t)} = \pi, (S_t, S_{t+1}) = (s_1, s_2)\}}{N_\pi(t)} \log \frac{J_{\pi_t}(\theta)}{J_{\pi_*}(\theta)}$$

$$= - \sum_{(s_1, s_2) \in \mathcal{S}^2} \tilde{\tau}_{t_k, k_\pi} \mathcal{H}_{(s_1, s_2)} (\tilde{\tau}_{t_k, k_\pi}, \pi) \log \frac{J_{\pi_t}(\theta)}{J_{\pi_*}(\theta)}$$

$$= \sum_{(s_1, s_2) \in \mathcal{S}^2} \left[ \tilde{\tau}_{t_k, k_\pi} \mathcal{H}_{(s_1, s_2)} (\tilde{\tau}_{t_k, k_\pi}, \pi) - k_\pi \tilde{\tau}_{t_k, k_\pi} \theta_\pi(s_1|s_2) \right] \log \frac{J_{\pi_*}(\theta)}{J_{\pi_t}(\theta)} + \sum_{(s_1, s_2) \in \mathcal{S}^2} k_\pi \tilde{\tau}_{t_k, k_\pi} \theta(s_1|s_2) \log \frac{J_{\pi_*}(\theta)}{J_{\pi_t}(\theta)} \tag{23}$$

The last equation is based on the logarithmic property $\log \frac{A}{B} = - \log \frac{B}{A}$. Based on the Assumption 3, define $\rho(x) := O(\sqrt{\log(\log(x))})$.

$$- \log W_{t_k}(\pi) \leq \sum_{(s_1, s_2) \in \mathcal{S}^2} \rho(k_\pi) \sqrt{k_\pi} \log \frac{J_{\pi_*}(\theta)}{J_{\pi_t}(\theta)} + k_\pi \tilde{\tau}_{t_k, k_\pi} \sum_{(s_1, s_2) \in \mathcal{S}^2} \theta(s_1|s_2) \log \frac{J_{\pi_*}(\theta)}{J_{\pi_t}(\theta)}$$

$$\leq \frac{\epsilon}{\Gamma} |S|^2 (\rho(k_\pi) \sqrt{k_\pi} + k_\pi \tilde{\tau}_{t_k, k_\pi}). \tag{24}$$

$\square$

**Lemma A.5.** *The difference between the local optimal average reward and the instantaneous average reward could be bounded by:*

$$|J^* - J_{\pi_t}| \leq \tilde{\mathcal{O}}(\frac{1}{\sqrt{T}}).$$

*Proof.* We could know that the current policy probability distribution is updated based on the previous distribution and the current local optimal policy distribution:

$$\mu_t(\pi) = W_t \mu_{t-1}(\pi) + (1 - W_t) \pi_t^*(s, \theta_{t_k}). \tag{25}$$

We could extend this result to reward function:

$$J_{\pi_t} = W_t J_{\pi_{t-1}} + (1 - W_t) J^*(\theta_t)$$

$$J_{\pi_t}^2 = W_t^2 J_{\pi t-1}^2 + (1 - W_t)^2 J^{*^2} + 2W_t (1 - W_t) J^* J_{\pi_{t-1}} \tag{26}$$

$$\leq W_t^2 J_{\pi t}^2 + (1 - W_t)^2 J^{*2} + 2W_t (1 - W_t) J^* J_{\pi_t}.$$

The inequality is based on the monotonicity of the algorithm. We could simplify Equation 26:

$$(1 - W_t^2) J_{\pi t}^2 \leq (1 - W_t)^2 J^{*^2} + 2W_t (1 - W_t) J^* J_{\pi_t}$$

$$(1 + W_t) J_{\pi t}^2 \leq (1 - W_t) J^{*^2} + 2W_t J^* J_{\pi_t}$$

$$J_{\pi_t}^2 + W_t J_{\pi t}^2 \leq J^{*^2} - W_t J^{*^2} + 2W_t J^* J_{\pi_t} \tag{27}$$

$$W_t \left( J_{\pi_t}^2 + J^{*^2} \right) \leq J^{*^2} - J_{\pi_t}^2 + 2W_t J^* J_{\pi_t}$$

$$J_{\pi_t}^2 + J^{*^2} \leq \frac{1}{W_t} \left( J^{*^2} - J_{\pi_t}^2 \right) + 2J^* J_{\pi_t}.$$

Based on the definition of the regret of each time step, we could deduce the bound of the instantaneous regret:

$$(J_{\pi t} - J^*)^2 = J_{\pi_t}^2 + J^{*^2} - 2J_{\pi t} J^*$$

$$\leq \frac{1}{W_t} \left( J^{*^2} - J_{\pi_t}^2 \right) + 2J^* J_{\pi_t} - 2J_{\pi_t} J^*$$

$$= \frac{1}{W_t} \left( J^{*^2} - J_{\pi_t}^2 \right) \tag{28}$$

$$= \frac{1}{W_t} (J^* - J_{\pi_t}) (J^* + J_{\pi_t}).$$

Therefore, we could deduce that:

$$|J_{\pi_t} - J^*| \le \frac{1}{W_t} |J_{\pi_t} + J^*|. \tag{29}$$

From Lemma A.4, we could know that $-\log W_{t_k}(\pi)$ is bounded by $B$, with $B = \frac{\epsilon}{\Gamma}|S|^2(\rho(k_\pi)\sqrt{k_\pi} + k_\pi \tilde{\tau}_{t_k, k_\pi})$. Therefore, we could construct the following inequalities.

$$W_{t_k} - 1 \ge \log W_{t_k} \ge -B$$
$$\frac{1}{W_{t_k}} \le \frac{1}{1 - B}. \tag{30}$$

Factor $B$ is proportional to parameter $k_\pi$ which could be bounded by the total number of episode of under total time $T$. Therefore, we could bound $\frac{1}{W_t}$ by $T$(Ignoring the constants):

$$\frac{1}{W_t} \le \frac{1}{1 - \frac{\epsilon}{\Gamma}|S|^2(\rho(k_\pi)\sqrt{k_\pi} + k_\pi \tilde{\tau}_{t_k, k_\pi})}$$
$$\le \frac{1}{1 - \sqrt{\sqrt{T}} - \sqrt{T}}. \tag{31}$$

Based on the definition of $\Gamma$, the average reward function is bounded by $\Gamma$. So the difference between the local optimal average reward and the instantaneous average reward could be bounded by:

$$|J^* - J_{\pi_t}| = |J_{\pi_t} - J^*|$$
$$\le \frac{1}{W_t} |J^* + J_{\pi_t}|$$
$$\le \frac{2}{1 - \frac{\epsilon}{\Gamma}|S|^2(\rho(k_\pi)\sqrt{k_\pi} + k_\pi \tilde{\tau}_{t_k, k_\pi})} \tag{32}$$
$$\le \tilde{\mathcal{O}}(\frac{2\Gamma}{S^2 \sqrt{T}}).$$

$\square$

**Lemma A.6.**

$$\frac{\Gamma}{S^2} < D\sqrt{SA}$$

*when $|S| \ge 2$ or $|A| \ge 2$*

*Proof.* We could know that $\Gamma$ is defined as the upper bound of the average reward. So we could deduct:

$$\Gamma \ge J_{\pi_*}(\theta_*)$$
$$D \ge \max T_{s \to s'}^{\pi t_k}(\theta_{t_k}) \tag{33}$$
$$\Gamma \le \max T_{s \to s'}^{\pi_*}(\theta_*) \le \max T_{s \to s'}^{\pi t_k}(\theta_{t_k}) \le D.$$

Assuming Lemma A.6 is established, we could get:

$$\frac{\Gamma}{S^2} \le \frac{D}{S^2} \le D\sqrt{SA}$$
$$D \le DS^2\sqrt{SA} \tag{34}$$
$$S^2\sqrt{SA} \ge 1.$$

Therefore the Lemma could be proved when the fMDP process has more than one state and one action. $\square$

**Theorem A.7.** *The first part of the regret in time step $T$ is bounded by:*

$$Reg_T^1 \le \tilde{\mathcal{O}}(\frac{\sqrt{T}}{S^2}).$$

*Proof.* From the definition before, we could know that $Reg_T^1$ could be represented as:

$$Reg_T^1 = TJ_{\pi_k}(\theta_K) - \sum_{t=1}^{T} J_{\pi_t}(\theta_K). \tag{35}$$

Since this theorem won't involve the transformation of the transition probability. So let $J_\pi(\theta) = J_\pi$. Based on the update rule of the posterior distribution $\mu_{t+1}(\pi)$ of policy $\pi$. We could divide the average reward into several parts:

At time step $t = T$, we could assume the instantaneous regret equals to zero:

$$Reg_{t_T}^1 = J_{\pi_k} - J_{\pi_T} = 0. \tag{36}$$

At time step $t = T - 1$, define the local optimal average reward as $J_\pi^*$. Note that this local optimal value is virtual. The instantaneous regret could be represented as:

$$\begin{aligned} Reg_{t_{T-1}}^1 &= J_{\pi_k} - J_{\pi_{T-1}} \\ &= W_{t-1}J_{\pi_{T-1}} + (1 - W_{t-1})J_\pi^* - J_{\pi_{T-1}} \\ &= (W_{t-1} - 1)J_{\pi_{T-1}} + (1 - W_{t-1})J_\pi^* \\ &= (1 - W_{t-1})(J_\pi^* - J_{\pi_{T-1}}). \end{aligned} \tag{37}$$

In a similar fashion, at time step $t = T - 2$, the instantaneous regret could be represented as:

$$\begin{aligned} Reg_{t_{T-2}}^1 &= J_{\pi_k} - J_{\pi_{T-2}} \\ &= J_{\pi_k} - J_{\pi_{T-1}} + J_{\pi_{T-1}} - J_{\pi_{T-2}} \\ &= (1 - W_{t-1})(J_\pi^* - J_{\pi_{T-1}}) + (1 - W_{t-2})(J_\pi^* - J_{\pi_{T-1}}). \end{aligned} \tag{38}$$

Based on Lemma A.5, the difference between the local optimal value and the current average reward could be bounded by:

$$|J_\pi^* - J_{\pi_t}| \leq \tilde{\mathcal{O}}(\frac{1}{\sqrt{T}}). \tag{39}$$

The sub-optimal models are sampled when their posterior probability is larger than $\frac{1}{T}$. This ensures the time complexity of the Thompson sampling process is no more than $O(1)$. So we could deduce the total regret in time step $T$.

$$\begin{aligned} Reg_T^1 &= \frac{1}{T}(Reg_{t_{T-1}}^1 + Reg_{t_{T-2}}^1 + \cdots + Reg_{t_1}^1) \\ &\leq \tilde{\mathcal{O}}(\frac{2\Gamma}{S^2\sqrt{T}})(\frac{T-1}{T} + \frac{T-2}{T} + \cdots \frac{1}{T}) \\ &\leq \tilde{\mathcal{O}}(\frac{2\Gamma\sqrt{T}}{S^2}). \end{aligned} \tag{40}$$

$\square$

In order to deduce the second regret bound generated by the transition probability, we should analyze our algorithm's performance over $T$ time step. We define the number of macro episodes $M = \mathbb{1}\{t_k \leq T\}$. An episode is defined as the set of the time steps under stopping criterions. Therefore, we could deduce the bound of the number of episode.

**Lemma A.8.** *Under the stopping criterion, the number of episodes $M$ could be bounded by:*

$$M \leq SA\log(T).$$

*Wei et al. (2017)*

*Proof.* The stopping criterion is triggered whenever the visits number of the initial state-action pair is doubled. So $M$ could be represented as:

$$M_{(s,a)} = \left\{ k \leq K_T : N_{t_k}(s,a) > 2N_{t_{k-1}}(s,a) \right\}. \tag{41}$$

Since the number of the visit to state-action pair $(s, a)$ is doubled at the beginning of every epoch $k$. The size of $\mathcal{M}_{(s,a)}$ should be no larger than $O(\log(T))$. Assume $\left|\mathcal{M}_{(s,a)}\right| \geq \log\left(N_{T+1}(s, a)\right) + 1$. We could have:

$$
\begin{aligned}
N_{t_{K_T}}(s, a) &= \prod_{k \leq K_T, N_{t_{k-1}}(s,a) \geq 1} \frac{N_{t_k}(s, a)}{N_{t_{k-1}}(s, a)} \\
&> \prod_{k \in \mathcal{M}_{(s,a)}, N_{t_{k-1}}(s,a) \geq 1} 2 \\
&\geq N_{T+1}(s, a).
\end{aligned}
\tag{42}
$$

This contradicts the fact that $N_{t_{K_T}}(s, a) \leq N_{T+1}(s, a)$. This leads to $\left|\mathcal{M}_{(s,a)}\right| \leq \log\left(N_{T+1}(s, a)\right)$. Therefore, we could obtain the bound of the number of the episodes:

$$
\begin{aligned}
M &\leq 1 + \sum_{(s,a)} \left|\mathcal{M}_{(s,a)}\right| \\
&\leq 1 + \sum_{(s,a)} \log\left(N_{T+1}(s, a)\right) \\
&\leq 1 + SA \log\left(\sum_{(s,a)} N_{T+1}(s, a)/SA\right) \\
&= 1 + SA \log(T/SA).
\end{aligned}
\tag{43}
$$

Since the logarithmic function is concave, we could simplify the inequality to:

$$
M \leq SA \log(T).
\tag{44}
$$

$\square$

**Lemma A.9.** *The total number of episodes of total time step $T$ could be bounded by:*

$$
K_T \leq \sqrt{2SAT \log(T)}.
$$

*Wei et al. (2017)*

*Proof.* Define macro episodes with start times $t_{n_i}$, $i = 1, 2, \cdots$ where $t_{n_1} = t_1$, we could have

$$
t_{n_{i+1}} = \min\left\{t_k > t_{n_i} : N_{t_k}(s, a) > 2N_{t_{k-1}}(s, a)\right\}.
$$

Let $\tilde{T}_i = \sum_{k=n_i}^{n_{i+1}-1} T_k$ be the length of the $i$th episode. Therefore, within the $i$th macro episode, $T_k = T_{k-1} + 1$ for all $k = n_i, n_i + 1, \cdots, n_{i+1} - 2$.

$$
\begin{aligned}
\tilde{T}_i &= \sum_{k=n_i}^{n_{i+1}-1} T_k \\
&= \sum_{j=1}^{n_{i+1}-n_i-1} \left(T_{n_i-1} + j\right) + T_{n_{i+1}-1} \\
&\geq \sum_{j=1}^{n_{i+1}-n_i-1} (j+1) + 1 = 0.5\left(n_{i+1} - n_i\right)\left(n_{i+1} - n_i + 1\right).
\end{aligned}
\tag{45}
$$

Consequently, $n_{i+1} - n_i \leq \sqrt{2\tilde{T}_i}$, for all $i = 1, \cdots, M$. From this property, we could obtain:

$$
K_T = n_{M+1} - 1 = \sum_{i=1}^{M} \left(n_{i+1} - n_i\right) \leq \sum_{i=1}^{M} \sqrt{2\tilde{T}_i}.
\tag{46}
$$

Based on Equation 46 and $\sum_{i=1}^{M} \tilde{T}_i = T$, we could get:

$$
K_T \leq \sum_{i=1}^{M} \sqrt{2\tilde{T}_i} \leq \sqrt{M \sum_{i=1}^{M} 2\tilde{T}_i} = \sqrt{2MT}.
\tag{47}
$$

Where the second inequality is based on Cauchy-Schwarz inequality. From Lemma A.8, we could know that the number of the macro episodes until time $T$ is bounded by $M \leq SA \log(T)$. Therefore, the lemma could be proved. □

**Theorem A.10.** *The regret caused by transition matrix sampling could be bounded by:*

$$Reg_T^2 \leq \tilde{O}(D(2SAT)^{\frac{1}{4}}).$$

*Proof.* In this theorem, we mainly focus on the difference between transition probability. From the previous definition of the Bellman iterator of the average reward, we could have:

$$
\begin{aligned}
J(\theta_K) + b(\theta_K, \pi, s) &= r(s, \pi) + \sum_{s'} \theta_K\left(s' \mid s, \pi\right) b\left(\theta_K, \pi, s'\right) \\
J\left(\theta_t\right) + b\left(\theta_t, \pi, s\right) &= r(s, \pi) + \sum_{s'} \theta_t\left(s' \mid s, \pi\right) b\left(\theta_t, \pi, s'\right).
\end{aligned}
\tag{48}
$$

The difference between the average reward under near-optimal transition probability and instantaneous transition probability could be represented as:

$$
\begin{aligned}
J(\theta_K) - J\left(\theta_t\right) = {} & b\left(\theta_t, \pi\right) - b(\theta_K, \pi) \\
& + \sum_{s', s}\left(\theta_K\left(s' \mid s, \pi\right) b\left(\theta_K, \pi, s'\right) - \theta_t\left(s' \mid s, \pi\right) b\left(\theta_t, \pi, s'\right)\right).
\end{aligned}
\tag{49}
$$

We could bound the first term with the largest difference between each state:

$$
\begin{aligned}
0 \leq b(\theta_K, \pi, s) - b\left(\theta_t, \pi, s\right) \leq sp(b(\theta)) \leq D \\
0 \geq b\left(\theta_t, \pi, s\right) - b(\theta_K, \pi, s) \geq -D.
\end{aligned}
\tag{50}
$$

Based on Equation 50, we could bound the second term in a similar way:

$$
\begin{aligned}
& \sum_{s'} \theta_K\left(s' \mid s, \pi\right) b\left(\theta_K, \pi, s'\right) - \theta_t\left(s' \mid s, \pi\right) b\left(\theta_t, \pi, s'\right) \\
& \leq D \sum_{s'}\left(\theta_K\left(s' \mid s, \pi\right) - \theta_t\left(s' \mid s, \pi\right)\right).
\end{aligned}
\tag{51}
$$

Then, we define the total transition difference as:

$$
\begin{aligned}
\theta_*(s'|s, \pi) - \theta_t(s'|s, \pi) &= \theta_K(s'|s, \pi) - \theta_t(s'|s, \pi) \\
\theta_k(s'|s, \pi) - \theta_t(s'|s, \pi) &= \theta_K(s'|s, \pi) - \theta_{K-1}(s'|s, \pi) + \theta_{K-1}(s'|s, \pi) - \cdots - \theta_t(s'|s, \pi).
\end{aligned}
\tag{52}
$$

From Equation 52, we could deduct the one-step transition difference to be:

$$
\theta_t(s'|s, \pi) - \theta_{t-1}(s'|s, \pi) = \frac{\theta_{t-1}(s'|s, \pi) + H\left(N_\pi(k), \pi\right)}{N_\pi(k)} - \theta_{t-1}(s'|s, \pi).
\tag{53}
$$

Based on Assumption 3, we could bound the one-step transition difference:

$$
\begin{aligned}
\sum_{s_1, s_2}\left(\theta_t - \theta_{t-1}\right) &= \sum_{s_1, s_2}\left[\frac{\theta_{t-1} + H_{s_1, s_2}}{N} - \theta_{t-1}\right] \\
&= \sum_{s_1, s_2} \frac{\theta_{t-1} + H_{s_1, s_2}}{N} - \sum_{s_1, s_2} \theta_{t-1} \\
&= \sum_{s_1, s_2} \frac{\theta_{t-1} + H_{s_1, s_2}}{N} - T_{s_1 \to s_2}^\pi\left(\theta_{t-1}\right) \\
&= \sum_{s_1, s_2} \frac{\theta_{t-1} + H_{s_1, s_2}}{N} - \tilde{\tau}_{t-1}.
\end{aligned}
\tag{54}
$$

where

$$\sum_{s_1,s_2} \frac{\theta_{t-1} + H_{s_1,s_2}}{N} \geq \frac{H_{s_1,s_2(k,\pi)}}{N}. \tag{55}$$

When we use stationary policy $\pi$ in epoch $k$, we could know that the count function of the policy $\pi$ should be less or equal to the total number of epochs. Therefore, we could deduct that:

$$\sum_{s_1,s_2} \frac{\theta_{t-1} + H_{s_1,s_2}}{N} \geq \frac{H_{s_1,s_2(k,\pi)}}{N} \geq \frac{H_{s_1,s_2(k,\pi)}}{k}. \tag{56}$$

Based on Assumption 3, we could deduct the bound for $\sum_{s_1,s_2}(\theta_t - \theta_{t-1})$:

$$\sum_{s_1,s_2}(\theta_t - \theta_{t-1}) \leq \sqrt{\frac{e_1 \log(e_2 \log k)}{k}}. \tag{57}$$

Combining Equation 57 with Equation 52. Since the update of the transition matrix only happens once in each epoch, we could deduct the difference between the periodic transition matrix and instantaneous transition matrix based on A.9:

$$\begin{aligned}
\theta_K(s' \mid s, \pi) - \theta_1(s' \mid s, \pi) &\leq K\sqrt{\frac{e_1 \log(e_2 \log k)}{k}} \\
&\leq \sqrt{k e_1 \log(e_2 \log k)} \\
&\leq \sqrt{\sqrt{2SAT \log T} e_1 \log(e_2 \log k)}.
\end{aligned} \tag{58}$$

Therefore, we could combine Equation 58 with Equation 51:

$$\begin{aligned}
\sum_{t=1}^{T} J(\theta_K) - \sum_{t=1}^{T} J(\theta_t) &\leq D\sqrt{\sqrt{2SAT \log T} e_1 \log(e_2 \log k)} \\
&\leq \tilde{O}(D\sqrt{\sqrt{2SAT}}).
\end{aligned} \tag{59}$$

$\square$

