# OpenReview forum: "Online Reinforcement Learning via Posterior Sampling of Policy"
_ICLR.cc/2023/Conference — Submitted to ICLR 2023_

### Official Review · Reviewer_craj · 2022-10-23

**Confidence:** 5
**Correctness:** 2
**Technical Novelty And Significance:** 2
**Empirical Novelty And Significance:** 2
**Recommendation:** 1

**Clarity, Quality, Novelty And Reproducibility:**

The paper requires significant proofreading. The results are very problematic to reproduce based on the high-level description only, without any code or data.

**Strength And Weaknesses:**

Strengths: the paper contains both theoretical analysis and empirical evaluation of the proposed algorithm.

Weaknesses: the paper requires significant proofreading to be followed. There are many typos and inconsistencies, and settings, assumptions,  and algorithm descriptions are unclear. The empirical evaluation results are hard to interpret, and will be problematic to reproduce as there was no code or data provided. Citations are rather random.

Examples: $\Gamma$ appears on page 2, not counting the abstract, but is only defined on page 3. Citation for 'online reinforcement learning' on page 1, first line of introduction, points to an article on a specific application of online RL, not particularly relevant to what follows. 3.1, problem setting, defines finite MDP and then refers to  'episodes' and 'epochs', which are not defined, and are used in a confusing way. Why the total reward of an episode should be in [0, 1]? How is this related to the reward function r to be SxA -> [0, 1]? In Assumption 3.3, it is not clear what do $e_1$ and $e_2$ correspond to, and this is not defined. In Algorithm 1, "game environment" suddenly pops up, without having been defined anywhere earlier. And so on, so forth.

**Summary Of The Paper:**

The paper proposes a sampling-based policy search for MDPs. The algorithm uses Thompson sampling over policies rather than over individual actions within policies. The algorithm is empirically evaluated and theoretically analyzed.

**Summary Of The Review:**

The paper may contain interesting ideas, however significant work is required to make it ready for publication.

---

> ### Author Response · Authors · 2022-11-17
> **Point-wise response to review**
>
> We thank Reviewer craj for the constructive comments on the weakness and provide our point-wise response below.
>
> ### Q1: The $\Gamma$ presentation problem of the paper
> Thank you for your advice on the presentation problem. We would like to introduce the definition of $\Gamma$ first in page 2. And elaborate its value connected to average reward in page 3.
>
> ### Q2: The citation problem
> Thank you for your advice on the presentation problem. We have cited papers that studied multiple aspects in order to give more general information to the readers.
>
> ### Q3: The confusion between epochs and episodes
> Thank you for your advice on the presentation problem. We are sorry for the confusion on epochs and episodes. We have changed all the 'epochs' to 'episodes'.
>
> ### Q4: Why the total reward of an episode should be in [0, 1]?
> Our method is defined under the finite-horizon MDP. It is an important assumption that the reward of the finite MDP would be normalized and constrained by [0,1]. The reward function $r: S \times A \rightarrow[0,1]^2$ is the mathcal representation for that.
>
> ### Q5: The confusion on $e_{1}$ and $e_{2}$
> This assumption is inherited from the previous paper[1]. They are simply two constants that satisfies $p \triangleq p\left(e_1, e_2\right)>0$. Which would give us the bound of the diference between empirical state pair frequency and corresponding expected value.
>
>
> **Reference**
> [1]Aditya Gopalan and Shie Mannor. Thompson sampling for learning parameterized markov decision processes. In Conference on Learning Theory, pp. 861–898. PMLR, 2015.

---

### Official Review · Reviewer_RWGK · 2022-10-24

**Confidence:** 4
**Clarity, Quality, Novelty And Reproducibility:** Please find my comments in the streng…
**Correctness:** 2
**Technical Novelty And Significance:** 1
**Empirical Novelty And Significance:** Not applicable
**Recommendation:** 1

**Strength And Weaknesses:**

While I acknowledge that both average-reward MDPs, as well as the posterior sampling type algorithms themselves are interesting objects in RL theory, I think overall I am highly concerned about the contribution of this paper, due to its lack of awareness of existing work, as well as the many clarity / consistency issues within the basic definitions and settings.

* For high-level comparisons of existing work, first of all, I think the authors did not make clear that they are restricting to the average-reward MDP setting, which is strange—There are at least two equally popular models, discounted MDPs, and finite-horizon MDPs, that are extremely well-studied in terms of online RL in the tabular case, but almost not mentioned at all in related work. With the exception of mentioning Jin et al. (2018) and Zhang et al. (2020), which the authors claim “have not achieved the state-of-art regret bound”. However, Jin et al.’s result is only one H (horizon) factor from optimal regret, and this was closed by Zhang et al. (2020)’s algorithm which achieves the minimax regret $\sqrt{H^3SAT}$ in the finite-horizon setting.

* For the regret bound, I think the current best result for average-reward MDPs is already $\sqrt{DSAT}$, for example, achieved by the work of Zhang and Ji (2019, Corollary 1). Thus the claim of the established $D\sqrt{SAT}$ result “matching the current best regret bound” sounds questionable.
[Reference] Zhang, Zihan, and Xiangyang Ji. "Regret minimization for reinforcement learning by evaluating the optimal bias function." Advances in Neural Information Processing Systems 32 (2019).

* Many basic definitions seem non-standard and inconsistent throughout the paper. For example, the definition (2) of regret does not look like the standard definition for RL (it maximizes over a single action $a$ instead of policies). Eq (2) also disagrees with the later appearance of Regret in Eq (10). Neither agrees with the standard definition of regret, which measures the difference against the optimal average reward (e.g. Zhang and Ji, Page 3).

* The authors define $\theta_{t_k}$ as the “transition probability” of the “model-based agents” within time-step $k$ and epoch $t$ (Section 3.1), whereas $\theta_\star$ is the true transition probability. What is the “transition probability of model-based agents”? Do the authors mean some estimate maintained within the algorithm?

* Assumption 3.1, why is “$\epsilon$-optimal policy” measured on this (estimated) $\theta_{t_k}$? I think the standard definition is to measure the deployed policy versus the optimal policy on the ground truth model $\theta_\star$?

Overall, because of these concerns, I am quite concerned about the contributions even before getting into the actual algorithms / analyses.



**Summary Of The Paper:**

This paper studies reinforcement learning theory in the average-reward MDP setting. The main result is a $\tilde{O}(D\sqrt{SAT})$ regret bound for a new algorithm that performs posterior sampling over policies.

**Summary Of The Review:**

This paper studies a topic that received wide attention in recent years (online RL in average-reward MDPs). However, the paper seems quite lacking in awareness of recent work, and has many issues in basic definitions and the clarity of the setups, which makes its contributions questionable in my opinion.

---

> ### Author Response · Authors · 2022-11-17
> **Point-wise response to review**
>
> We thank Reviewer RWGK for the constructive comments on the weakness and provide our point-wise response below.
>
> ### Q1: The relationship between our paper and average-reward MDP setting
> Thanks for pointing out the average-reward MDP. Our setting do have a lot in common with average-reward MDP. First of all, our method is conducted under the finite-horizon MDP which had been shown in Section 3.1. The average-reward MDP was first defined in[1]. The algorithm for average-reward MDP focus on the optimization of the average reward which matches our paper's setting. Second, the model-free algorithm that had been brought up in Jin et al.(2018) had a regret bound of $\tilde{\mathcal{O}}\left(\sqrt{H^3 S A T}\right)$ where $H$ matches $D$. Which do not match state-of-art regret bound $\tilde{\mathcal{O}}(D \sqrt{S A T})$.
>
> ### Q2: The question about the state-of-art method
> In Zhang and Ji(2020)[2], they proposed a model-free method that achieves. They also claimed that 'Our regret bound improves upon the results of [Jin et al., 2018] and matches the best known model-based algorithms as well as the information theoretic lower bound up to logarithmic factors.' As for the 2019 paper[3], it brought up strong assumption on the sampling method and achieve the sub-optimal regret bound $\tilde{O}(\sqrt{S A D T})$ which consider a more relax situation than our method.
>
> ### Q3: The non-standard representation
> The definition of the regret is represented by the difference of the actual total reward and expected total reward. These reward could be divided into each time step. When it is considered under the episodic scenario, it should be maximized over policies. When it is considered under each time step, it would be maximized over actions. As for the standard definition, the definition of the regret in Equation(10) could be achieved when we divide the total regret into posterior policy regret and posterior transition probability regret.
>
> ### Q4: The confusion about $\theta_{t_{k}}$
> We apologize for the confusion about the definition about $\theta_{t_{k}}$. The transition probability is learned throughout the learning process of the agent.
>
> ### Q5: why is $\epsilon$-optimal policy measured on $\theta_{t_{k}}$?
> As we mentioned in Section 3.1, the ground truth model $\theta_{*}$ can not be observed by the agent. So the agent has to learn the estimated transition model and $\theta_{t_{k}}$ represents the learned value of the agent in time step $t$ under episode $k$. We could only find the local optimal policy on the instantaneous estimated transition probability $\theta_{t_{k}}$. So we use the $\epsilon$-optimal policy to represent our local optimal policy. It is often extremely hard for the agent to achieve true optimal game velue. Therefore, we use the $\epsilon$-optimal metric to help tight our bound. When the agent learns the $epsilon$-optimal policy. We could assume that it achieves the optimal value.
>
> **Reference**
> [1]Mahadevan S. Average reward reinforcement learning: Foundations, algorithms, and empirical results[J]. Machine learning, 1996, 22(1): 159-195.
> [2]Zhang Z, Zhou Y, Ji X. Almost optimal model-free reinforcement learningvia reference-advantage decomposition[J]. Advances in Neural Information Processing Systems, 2020, 33: 15198-15207.
> [3]Zhang Z, Ji X. Regret minimization for reinforcement learning by evaluating the optimal bias function[J]. Advances in Neural Information Processing Systems, 2019, 32.

---

### Official Review · Reviewer_5tze · 2022-10-25

**Confidence:** 1
**Correctness:** 2
**Technical Novelty And Significance:** 3
**Empirical Novelty And Significance:** Not applicable
**Recommendation:** 3

**Clarity, Quality, Novelty And Reproducibility:**

In the current state, the submission is really hard to read, see specific comments. Many notations are inconsistent of absent.  Some sentences are not finished or make no sense. Therefore, I'm not able to provide any feedback on the core ideas of this submission.

#Specific comments:
- P1, Abstract: What do you mean by 'gain the globally sub-optimal rewards'?
- P1, Section 1: I'm not sure that Kevton et al. 2020 is the only paper dealing with the exploration-exploitation dilemma in reinforcement learning.
- P1, Section 1: The references Ding et al., 2021;Oh & Iyengar, 2019; Moradipari et al., 2019 concerns rather contextual bandits problems than reinforcement learning problem, maybe you could consider citing, e.g.  Agrawal and Jia (2017b),Osband and Van Roy, 2017, Russo 2019.. who analyses Thompson sampling like algorithm for reinforcement learning.
- P1, Section 1: The statement 'The general optimistic algorithms require to solve all MDPs lying within the confident sets' is wrong. Take for example the UCBVI algorithm by Azar et al., 2017  that only needs to solve one MDP (with bonuses added to the rewards).
- P1, Section 1: What do you mean by 'Thompson sampling [..] results in biased estimates of the transition matrix' exactly?
- P1, Section 1: It is not clear for me how a 'Bayesian method to update transition probabilities' can have a regret bound?
- P1, Section 1: D is not defined. For the bound to be relevant you should precise the setting you are considering, e.g. infinite-horizon average reward...
- P1, Section 1: You need to define what you mean by  'samples posterior policy distributions' because for example in PSRL by Osband et al., 2013; algorithm by also a posterior policy is sampled (a policy from the posterior over optimal policy exactly which is obtained by sampling a posterior over the MDP and then computing the optimal policy of this particular MDP).
- P2, Section 2: What do you mean by 'total regret bound' exactly?
- P3, Section 3.1: 'probability could be defined The transition probability' sentence not finished. There is a repetition in the sentence after.
- P3, Section 3.1: What is epoch k?
- P3, Section 3.1: What is $\pi_{\theta_k} exactly? The globally optimal policy for which criteria? What is the 'local optimal transition probability'?
- P3, Section 3.1: How are generated the action a_t' in (1)?
- P3, Section 3.1: It is the maximum over what in the definition of \Gamma?
- P3, Section 3.1: How are generated the states s_t and action a_t in (2)?
- P3, Section 3.1, (2): You change notation for the expectation.
- P3, Section 3.1: It is a sum over what in the definition of r(s, \pi)
- P3, Section 3.1, definition of D: It is a maximum over what, it should not be \pi instead of \pi_{t_k}?
- P4, Section 3.2: What do you mean by 'online settings'?
- P4, Section 3.2: I do not understand the sentence 'As they often get stuck in locally optimal results' which results?
- P4, Section 3.2, Assumption 3.1: The quantity J_\pi(\theta) is not defined you only introduced J(\pi). It seems to be a definition.
- P4, Section 3.2: You still not defined properly the transition time.
- P4, Section 3.2,Assumption 3.2: The max are not well defined and does this assumption holds for all t_k?
- P4, Section 3.2: What is the definition of N(\pi) exactly?
- P4, Section 3.2: The posterior probability under which Bayesian model?
- P5, Section 4: Since you did not define the Bayesian model and \mu in particular it is complicated to follow the presentation of Algorithm 1.

**Strength And Weaknesses:**

Weaknesses:

- Presentation

**Summary Of The Paper:**

The authors consider the regret minimization problem for infinite-horizon average-reward Markov decision process (MDP). They propose the Reward-Weighted Posterior Sampling of Policy (RWPSP) a Thompson sampling inspired algorithm. Precisely  RWPSP maintains a posterior distribution over policies with is updated with the optimal policy of the estimated probability transitions. They prove a regret bound of order O(\Gamma \sqrt(T)/S^2) where \Gamma /S^2\leq D\sqrt{SA} where T is the horizon, S the number of states, A the number of actions and D the diameter of the MDP. They experimentally show that  RWPSP compares favorably to other baselines.

**Summary Of The Review:**

See above

---

> ### Author Response · Authors · 2022-11-17
> **Point-wise response to the review**
>
> We thank Reviewer 5tze for the constructive comments on the weakness and provide our point-wise response below.
>
> ### Q1: What is the meaning of 'gain the globally sub-optimal rewards'
> We meant to show that only sampling the transition probablity could not achieve the global optimal rewards for the MDP. But there is no gurantee for every algorithm to achieve the global optimal rewards, so we decided to use the globally sub-optimal rewards instead.
>
> ### Q2: Is Kevton et al. 2020 is the only paper dealing with the exploration-exploitation dilemma in reinforcement learning?
> There are many of the researchers that tried to propose the solution of the exploration-exploitation(EE) dilemma in RL. We chose this paper as an example since it matches our game setting. We have added more papers that concern the EE dilemma.
>
> ### Q3: Consideration in citing references concerns contextual bandits
> Thank you for your valuable advices. We have added the reference in our paper in Section 1.
>
> ### Q4: The correctness of the statement 'The general optimistic algorithms require to solve all MDPs lying within the confident sets'
> In this sentence, we simply consider the differences between the Thompson sampling-based algorithms and other general optimistic algorithms. General UCB-based algorithms like UCBVI construct confidence set for the MDP and solve the MDP with self-updated confidence set. We haven't expressed our opinion clearly in this sentence. It should be corrected as: 'The general optimistic algorithms require to solve multiple confident set on the MDP'.
>
> ### Q5: The meaning of 'Thompson sampling [..] results in biased estimates of the transition matrix'
> The idea that we are trying to express is that, though Thompson sampling method speed up our learning process of the transition probability. We should be aware that such approximation actually trade time with preciseness. The sampled transition matrix would be biased.
>
> ### Q6: How a 'Bayesian method to update transition probabilities' can have a regret bound?
> Bayesian method like Thompson sampling-based method tried to acquire the optimal game value by updating the predicted transition probability and policy distribution. During that process, previous work introduced regret to measure the performance of such methods. The definition of the regret is presented in Section 3.1, Equation(2). It is defined as the difference between the optimal total game value and the actual game value. Its upper bound shows how much the learned policy close to the optimal policy.
>
> ### Q7: The lackness of the definition of D
> We shown in the first sentence of Section 3.1 that our paper considers the finite stochastic Markov decision process. And the definition of D is also presented in the Section 3.1 in the last sentence. Which is the maximum expected time to reach to state $s^{\prime}$ from state $s$ under transition probability $θ_{t_{k}}$
> and policy $π_{t_{k}}$.
>
> ### Q8: The confusion about 'samples posterior policy distributions'
> We apologize for the confusion. In our paper, we construct a policy distribution which is the posterior sampled from the previous policy distribution. What PSRL did was to sample the posterior transition probability of the MDP and then planning based on this transition function. While We focus on the policy itself and sample from the policy distribution directly. Detailed description has been added in the paper.
>
> ### Q9: The meaning of total regret bound
> The formal definition of the total regret bound is appeared in Section 5.2, Equation(10). We use the total regret bound in order to analyze the posterior policy regret $Reg^{1}$ and posterior transition probability regret $Reg^{2}$ respectively.
>
> ### Q10: The incompleteness of the sentence
> We apologize for the incompleteness of the sentence, the proper sentence should be 'For the model-based agents, the transition probability at time step $t$ within epoch $k$ could be defined as $θ_{t_{k}}$.' The sentence had been corrected.
>
> ### Q11: What is epoch k?
> We apologize for the mistake of the definition of k. K would represent the episode number. We have corrected it in our paper.
>
> ### Q12: What is $\pi_{\theta_k}$?
> In our paper, we assume that the agent would perform better under better-learned transition function and policy. $\pi_{\theta_k}$ is defined as the policy the agent would conduct under its learned transition function would be $\theta_{k}$. The globally optimal policy is the true optimal policy for the MDP. The agent would gain the optimal game value when they conduct globally optimal policy. As for the local optimal transition probability, it is learned through each episode. When the agent met our stopping criterion for a single episode, we could safely assume that the local optimal transition probability is learned.

---

> > ### Author Response · Authors · 2022-11-17
> > **Continued Part**
> >
> > ### Q13: How are generated the action $a_{t^{\prime}}$ in (1)?
> > At the beginning of the algorithm, the action would be randomly generated in order to gather more data. When it meets the criterion we set, the action would be sampled from the posterior policy distribution. Detailed description had been added in the paper.
> >
> > ### Q14: It is the maximum over what in the definition of \Gamma?
> > We apologize for the unclearness for the definition of $\Gamma$. $\Gamma$ is defined as the maximum average reward. Its value varies based on the learned transition matrix as well as the policy agent conducts. Therefore, $\Gamma$ would achieve its maximum value when the agent conducts optimal policy under optimal learned transition matrix.
> >
> > ### Q15: How are generated the states s_t and action a_t in (2)?
> > We discuss our algorithm under online learning setting. Which means the states $s_{t}$ and action $a_{t}$ would be generated from the interaction between the agent and environment.
> >
> > ### Q16: The notation for the expectation in Equation(2)
> > In the definition of the regret bound, there would be no involvement of the expectation. The expectation only exists in the Bayesian regret bound.
> >
> > ### Q17: What would be sum over in the definition of $r(s,\pi)$
> > The $r(s,\pi)$ represents the expected
> > total reward under stationary policy $\pi$. So the expecation would be sum over the policy $\pi$
> >
> > ### Q18：What is the maximum over D?
> > $D$ is defined as the maximum expected time to reach to state $s^{\prime}$ from state $s$ under transition probability $\theta_{t_{k}}$ and policy $π_{t_{k}}$. Under such circumstance, we believe $\pi_{t_{k}}$ would be more appropriate. We discussed the definition of $D$ in the Assumption 3.2 specifically.
> >
> > ### Q19: What do you mean by 'online settings'?
> > In the online learning setting, there are two major features: First, all the training data should be acquire by the interaction process between the agent and the environment. Second, the measurement of the performance of the online learning algorithms is usually regret bound. Such bound would strongly correlated with the total time. Which means the convergence proof and the upper bound would be important in the online learning setting.
> >
> > ### Q20: I do not understand the sentence 'As they often get stuck in locally optimal results' which results?
> > During the exploration of the agent, it is possible that the local optimal value does not fit with the global optimal value. So some of the agents would stuck in the locally optimal results.
> >
> > ### Q21: The lack of definition of $J_{\pi}(\theta)$
> > We apologize for the lack of definition of $J_{\pi}(\theta)$. It would be the average reward when agent conduct policy $\pi$ under learned transition matrix $\theta$. We have added in the paper.
> >
> > ### Q22: Lack of the definition of the transition time
> > The transition time is defined as the expected time to reach to state $s^{\prime}$ from state $s$
> >
> > ### Q23: The max are not well defined and does this assumption holds for all t_k?
> > The max are defined on the total transition time T and this assumption do hold for every $t_{k}$ since it consider the worst case.
> >
> > ### Q24: What is the definition of N(\pi) exactly?
> > Let $e(t) = k$ be the epoch where time instant $t$ belongs. When conducting stationary policy $\pi$, we could define the count function for the episode number $e(t)$ as $N(\pi_{e(t)})$.
> >
> > ### Q25: The posterior probability under which Bayesian model?
> > In our paper, we construct the posterior probability on the transition model and policy distribution.
> >
> > ### Q26: The lack of clearity of the definition of the Bayesian model and $\mu$
> > We apologize for the lack of clearity. $\mu$ is represented as the posterior distribution. And $\mu(\pi)$ would represent the posterior policy distribution. We have further elaborated the definition of the $\mu$ in the paper.

---

### Official Review · Reviewer_j9Gs · 2022-11-04

**Confidence:** 3
**Correctness:** 2
**Technical Novelty And Significance:** 3
**Empirical Novelty And Significance:** 2
**Recommendation:** 3

**Clarity, Quality, Novelty And Reproducibility:**

The paper seems well structured at a high level, but once I started looking into the details it is severly lacking. I made a fair attempt at understanding the proof, but due to incomplete notation and very barebones derivations in some places I would not be confident in its meaningfulness.

The quality of the paper is negatively impacted by the lack of clarity.

Similar, if there are novel contributions beyond the algorithmic one they should be appropriately highlighted in the proof. Which part of the derivation is made possible by sampling directly posterior policies rather than full transition models?

The reproducibility of the theoretical part is not very good due to lack of clarity. The experimental part is not reproducible, as the authors do not report any of the relevant hyperparameters to run RWPSP, nor include code.



**Strength And Weaknesses:**

The main strength of the method are matching (and potentially improving) the current best rates for this setting, the novelty of the approach to the posterior sampling family, and the seemingly complete lack of propabilistic statements in the regret proof.

The weaknesses are:
- The claim of optimality of O(D(SAT)^1/2) regret only holds for posterior sampling approaches (the paper "Regret Minimization for Reinforcement Learning by Evaluating the Optimal Bias Function" Zihan Zhang, Xiangyang Ji, NeurIPS2019 seems to report a O((DSAT)^1/2) bound under comparable assumptions). I am not sure if since 2019 no other posterior sampling approach also matched this bound.
- Could the authors clarify the proof of lemma 5.8? From Lemma A.6 in the appendix we have equation (33) proving \Gamma \leq D by application of Assumption 3.2. At this point further bounding D with DS^2(SA)^1/2 seems pointless. A finite MDP cannot have less than 1 state, so a direct application of \Gamma \leq D would yield a O(DT^1/2/S^2) regret, much better than existing results.
- Lemma A.6 seems to rely on Assumption 3.2 heavily. However I do not see why it would be natural to assume that the optimal policy would reduce the diameter between *any two states* in the MDP compared to any of the policies encountered during optimization. In particular the MDP reward (and therefore the optimal policy) does not usually care about arbitrary transition times, but only about transitioning to a good state as quickly as possible. On the other hand a sub-optimal policy might be well aligned (due to sheer randomness) with the explorative goal of tansversing the whole space as efficiently as possible.
- Assumption 3.2 does not define what would the max be over.
- I am not fully familiar with the setting, but the complete lack of any probabilistic statement in any of the results (i.e. none of the theorem hold only with some probability, and the regret bounds seem to rely only on the assumptions). The only statistical comment is a brief reference in section 5.2. This seems very unlikely to be rigorous considering posterior sampling is a randomized algorithm. If this was indeed the case it would be a significant aspect of the contribution and should be highlighted in the text.
- The exposition of the whole paper is well structured, but clarity falls apart at a more careful inspection. Most of the maxes are missing the variable that are being maxed on. Section 3.1 contains broken phrases, including crucially the one that introduces \theta. p_\theta is never introduced. the span is mentioned once and never again. Lemma 5.3 uses a different notation in the main paper and in the appendix.

**Summary Of The Paper:**

The paper introduces a new posterior sampling algorithm (RWPSP) for the horizon-free finite state tabular MDP setting in RL. The algorithm is part of the posterior sampling family of approaches (e.g. a la Thompson sampling), but differentiates itself from past results by:
- not attempting to sample an estimate the transition model and optimize a policy out of that
- but instead building directly a posterior on policies based on an estimate transition model, and then sample a policy out of that

This different approach results in a different regret bound of O(\Gamma T^1/2 S^-2) which the authors show is always smaller or equal than the current best bounds of O(D(SAT)^1/2).

Finally, preliminary results on toy MDPs are provided to highlight the improved performance in terms of regret of the proposed method.

**Summary Of The Review:**

Overall I feel the paper introduces a nice idea (side-stepping the problem of estimating and sampling the transition model), but I could not understand from the theoretical derivation how this change unlocks new solutions in the regret analysis. Due to this lack of clarity and confusion in the soundness of the result I would not reccomend acceptance for the paper in its current state.

However, despite having a background in theoretical RL, I also recognize that the field is extremely specialized and I might be missing necessary context and conventions given for granted in the proof derivation, and remain open to re-evaluate my position if the authors can clarify their work.

---

> ### Author Response · Authors · 2022-11-17
> **Point-wise response for the questions**
>
> We thank Reviewer j9Gs for the constructive comments on the weakness and provide our point-wise response below.
>
> ### Q1: Is there other posterior sampling approach that matches the bound?
> There are a number of algorithms that included posterior sampling after the year 2019.
>
> - Posterior Sampling algorithms that proposed after 2019
>
> | Algorithm          | Upper Regret Bound               |
> | ----------------- |:-----------------------: |
> | UCB-Advantage[1]       | $\widetilde{\mathcal{O}}\left(\sqrt{H^3 S A T}+H^{33 / 4} S^2 A^{3 / 2} T^{1 / 4}\right)$   |
> | UCBMQ[2] | $\widetilde{\mathcal{O}}\left(\sqrt{H^3 S A T}+H^4 S A\right)$     |
> | MVP[3]         | $O\left(\left(\sqrt{S A K}+S^2 A\right)\right.$ poly $\left.\log (S A H K)\right)$     |
> | Horizon-Free Reinforcement Learning[4]        | $O\left(\left(\sqrt{S^9 A^3 K}\right)\right.$ polylog $\left.(S, A, K)\right)$    |
> | RLSVI[5]   | $\widetilde{\mathcal{O}}\left(H^2 S \sqrt{A T}\right)$ |
> |Near-optimal Randomized exploration[6]|$\widetilde{O}(H \sqrt{S A T})$|
>
> In the above presenting algorithms, H is the diameter of the MDP which possesses the same meaning as parameter D in our paper. These algorithms had not shown significant improvement comparing to our approach.
>
> ### Q2: Clarity of the proof of Lemma 5.8
> In Appendix A.6, we first assumed that this Lemma is sound in the first part. Then we could use the following Equation(34) to prove the Lemma inversely.
>
> ### Q3&Q4: Question about the Assumption 3.2
> In the traditional background of such posterior sampling approach, algorithms are required to learn and predict the transition probability of the MDP. In our paper, we include the consideration of the policy distribution in Assumption 3.2. We discuss the maximum expected time to reach to state $s^{\prime}$ under sub-optimal predicted transition functions and/or sub-optimal policy.
>
> The max would be over the transition time T since we are considering the maximum transition time. Therefore, as we are considering the maximum transition time, sub-optimal policy under sub-optimal predicted transition probability would suffer for randomness and waste more time on exploring useless states. The same assumption could be applied to the scenario when agent apply optimal policy with sub-optimal transition probability.
>
>
> ### Q5: The lackness of probabilistic statement in regret bound
> In the proof of the regret bound, we always consider for the worst case. This would reflect on the upper regret bound. The randomness of the algorithm is reduced the minimum level in order to get fair measurement of the performance of the algorithms. We will add up the relevant description in our paper, thanks for the advice.
>
> ### Q6: The clarity of the paper
> We used $\theta$ as the denotation of the transition probability in Section 3.1. The definition of span was appeared in section 3.1 and it is upper bounded by diameter D. Which we introduced in Section 3.1 just under Equation(4). Appendix A.2 has falsely used $v_{i}$, it should be $u_{i}$ instead. Thank you for your correction.
>
> **Reference**
> [1]Zhang Z, Zhou Y, Ji X. Almost optimal model-free reinforcement learningvia reference-advantage decomposition[J]. Advances in Neural Information Processing Systems, 2020, 33: 15198-15207.
> [2]Ménard P, Domingues O D, Shang X, et al. UCB Momentum Q-learning: Correcting the bias without forgetting[C]//International Conference on Machine Learning. PMLR, 2021: 7609-7618.
> [3]Zhang Z, Ji X, Du S. Is reinforcement learning more difficult than bandits? a near-optimal algorithm escaping the curse of horizon[C]//Conference on Learning Theory. PMLR, 2021: 4528-4531.
> [4]Zhang Z, Ji X, Du S. Horizon-free reinforcement learning in polynomial time: the power of stationary policies[C]//Conference on Learning Theory. PMLR, 2022: 3858-3904.
> [5]Agrawal P, Chen J, Jiang N. Improved worst-case regret bounds for randomized least-squares value iteration[C]//Proceedings of the AAAI Conference on Artificial Intelligence. 2021, 35(8): 6566-6573.
> [6]Xiong Z, Shen R, Du S S. Randomized exploration is near-optimal for tabular mdp[J]. arXiv preprint arXiv:2102.09703, 2021.

---

### Decision · Program_Chairs · 2023-01-20

**Decision:**

Reject

**Justification For Why Not Higher Score:**

Concerns about the writing and the comparison with prior work.

**Justification For Why Not Lower Score:**

N/A

**Metareview: Summary, Strengths And Weaknesses:**

This paper presents new posterior sampling methods for the average-reward setting. However, reviewers raised concerns about the writing and the comparison with prior work. The AC agrees and recommends rejection.